# Addressing people's current and future states in a reinforcement learning algorithm for persuading to quit smoking and to be physically active

Nele Albers[1]*, Mark A. Neerincx[1,2], Willem-Paul Brinkman[1]

1 Department of Intelligent Systems, Delft University of Technology, Delft, The Netherlands, 2 Department of Perceptual and Cognitive Systems, Nederlandse Organisatie voor Toegepast Natuurwetenschappelijk Onderzoek (TNO), Soesterberg, The Netherlands

* E-mail: n.albers@tudelft.nl

**Data Availability Statement:** The data are publicly available with the following DOI: https://doi.org/10.4121/21533055.v2 (This is the link to the data: https://data.4tu.nl/articles/dataset/Addressing_

## Abstract

Behavior change applications often assign their users activities such as tracking the number of smoked cigarettes or planning a running route. To help a user complete these activities, an application can persuade them in many ways. For example, it may help the user create a plan or mention the experience of peers. Intuitively, the application should thereby pick the message that is most likely to be motivating. In the simplest case, this could be the message that has been most effective in the past. However, one could consider several other elements in an algorithm to choose a message. Possible elements include the user's current state (e.g., self-efficacy), the user's future state after reading a message, and the user's similarity to the users on which data has been gathered. To test the added value of subsequently incorporating these elements into an algorithm that selects persuasive messages, we conducted an experiment in which more than 500 people in four conditions interacted with a text-based virtual coach. The experiment consisted of five sessions, in each of which participants were suggested a preparatory activity for quitting smoking or increasing physical activity together with a persuasive message. Our findings suggest that adding more elements to the algorithm is effective, especially in later sessions and for people who thought the activities were useful. Moreover, while we found some support for transferring knowledge between the two activity types, there was rather low agreement between the optimal policies computed separately for the two activity types. This suggests limited policy generalizability between activities for quitting smoking and those for increasing physical activity. We see our results as supporting the idea of constructing more complex persuasion algorithms. Our dataset on 2,366 persuasive messages sent to 671 people is published together with this article for researchers to build on our algorithm.

people_s_current_and_future_states_in_a_
reinforcement_learning_algorithm_for_
persuading_to_quit_smoking_and_to_be_
physically_active_Data_and_analysis_code/
21533055.

**Funding:** This work is part of the multidisciplinary research project Perfect Fit, which is supported by several funders organized by the Netherlands Organization for Scientific Research (NWO, https://www.nwo.nl/), program Commit2Data - Big Data & Health (project number 628.011.211). The funders had no role in study design, data collection and analysis, decision to publish, or preparation of the manuscript.

**Competing interests:** The authors have declared that no competing interests exist.

## Introduction

Imagine a woman called Janine who wants to motivate her friend Martha to become more physically active. Janine could motivate Martha to go for a run because that has worked for her other friends. However, this likely only works if Martha has running shoes. If she does not have any, just asking Martha to go for a walk might be more successful. So the success of the motivation may depend on the state Martha is currently in. In addition, if Janine cares about the overall success of all her motivational attempts, she should probably begin by telling Martha how to buy running shoes. This may cause Martha not to work out this week, but future attempts to motivate her to work out are much more likely to be successful once Martha has running shoes. So Janine should also consider the future states of Martha. And, lastly, people differ in whether they prefer to walk or run, no matter if they have running shoes. So Janine should also consider what type of person Martha is. Since Janine is not always available to motivate Martha, we want to create a virtual coach. Can this virtual coach do what Janine does?

Changing behavior such as becoming more physically active is crucial to improving health and reducing premature death. For example, 40% of deaths in the United States are brought about by unhealthy behavior [1, 2]. In addition, changing one behavior can make changing another one easier. For instance, becoming more physically active may facilitate quitting smoking [3, 4] and vice versa [5]. However, while many people want to change their behavior, doing so without help can be difficult. For example, more than two-thirds of adult smokers in the United States want to quit smoking [6], but most unassisted quit attempts fail [7]. One promising way to support people in changing their behavior are eHealth applications [8], which provide elements of healthcare over the Internet or connected technologies such as apps and text messaging. However, while such applications can be easy to use, available at all times, scalable, cost-effective, and can facilitate tailoring [9], adherence to them remains low [10, 11]. Adherence refers to whether and how thoroughly people do the activities suggested by the application.

We, therefore, aim to develop persuasion algorithms that successfully encourage people to adhere to their behavior change intervention. A one-size-fits-all approach to persuasion is unlikely to be effective [12, 13], as behavior change theories [14, 15] suggest many factors that affect personal behavior. However, these factors can be used as a starting point for designing algorithm-driven persuasion. Algorithm-driven persuasion is persuasion that is determined by programming code, with the advantage that it can use behavioral user data, target individuals or groups, and be adaptive [16]. Previous work on persuasion algorithms has shown that one can use data gathered on other people [17, 18], similar people [19, 20] or a single individual [17, 18, 21–24] to choose a persuasion type (e.g., advice from peers vs. experts). However, it is essential also to consider the context of a persuasive attempt [25–27]. One way to define a context is by describing the current state a persuadee is in. For example, Bertolotti et al. [28] show that the success of different messages to reduce red meat consumption depends on the persuadee's self-efficacy. In addition, persuasion types depend not only on the persuadee's state for their success, but they in turn also influence the persuadee's state for future persuasive attempts. For instance, messages for quitting smoking differ in their impact on self-efficacy [29]. Thus, if we want to maximize the effectiveness of persuasive attempts over time, we need to consider both current and future states.

One framework that allows us to formulate an adaptive and data-driven algorithm that considers both current and future states is Reinforcement Learning (RL). There are first results for applying RL with consideration of people's states to adapting the framing of messages for inducing healthy nutritional habits [30] or the affective behavior of a social robot teacher for

children [31]. In our approach, we investigate whether states are also helpful in persuading people to do preparatory activities for quitting smoking, such as writing down and ranking reasons for quitting smoking. In addition, we go a step further by also taking the similarity of people into account. The reason is that previous work has shown that characteristics such as the stage of behavior change [20] and personality [20, 32–38] affect the effectiveness of different persuasion types. The result is a personalized RL algorithm for choosing persuasive messages.

To systematically assess the value of subsequently adding the consideration of states, future states, and the similarity of people, we conducted a longitudinal experiment. Since the effects of these algorithm elements are difficult to assess in a complete behavior change intervention in which many other components such as goal-setting and progress feedback can play a role (e.g., see Brinkman et al. [39] in the context of usability testing), we created a minimal intervention in which people were only coached to *prepare* for changing their behavior. In this intervention, a conversational agent served as a virtual coach that suggested and persuaded people to do preparatory activities for quitting smoking. Since becoming more physically active may facilitate quitting smoking [3, 4], half of the activities addressed preparing for increasing physical activity.

## Hypotheses

The objective of this study was to test a personalized RL approach to persuading people to do preparatory activities for quitting smoking and increasing physical activity. The complete algorithm considers a person's state, future states, and the similarity of people when choosing a persuasion type. The goal thereby is that people do their activities more thoroughly, which is supposed to facilitate quitting smoking (Fig 1). Therefore, our first hypothesis is that subsequently incorporating the elements of the personalized RL algorithm is more effective with respect to how thoroughly people do their activities. Furthermore, our algorithm does not distinguish between preparatory activities for quitting smoking and ones for increasing physical activity because both types of activities serve the same behavioral goal of quitting smoking. This leads to our second hypothesis, which is that the best persuasion strategy is similar if we use data collected on both types of activities compared to using data collected on solely one type of activity. We now motivate each hypothesis in turn.

### H1: Algorithm effectiveness

In the introductory example, Martha wanted to become physically active. In the simplest case, the virtual coach could send her the persuasion type that has led people to do their activities most thoroughly in the past. As a measure for this thoroughness, the virtual coach could use the self-reported effort people put into assigned activities. Of course, if no such data is available yet, the virtual coach would need to choose either randomly or based on other sources of information such as experts due to the cold-start problem. Assuming such data is available and the effort for three persuasion types is as shown in Fig 2A, for instance, the virtual coach would

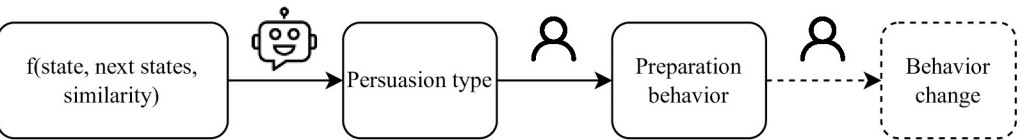

**Fig 1. Overarching goal of our work.** The goal of our persuasion algorithm is that people do their preparatory activities more thoroughly, which is supposed to facilitate quitting smoking.

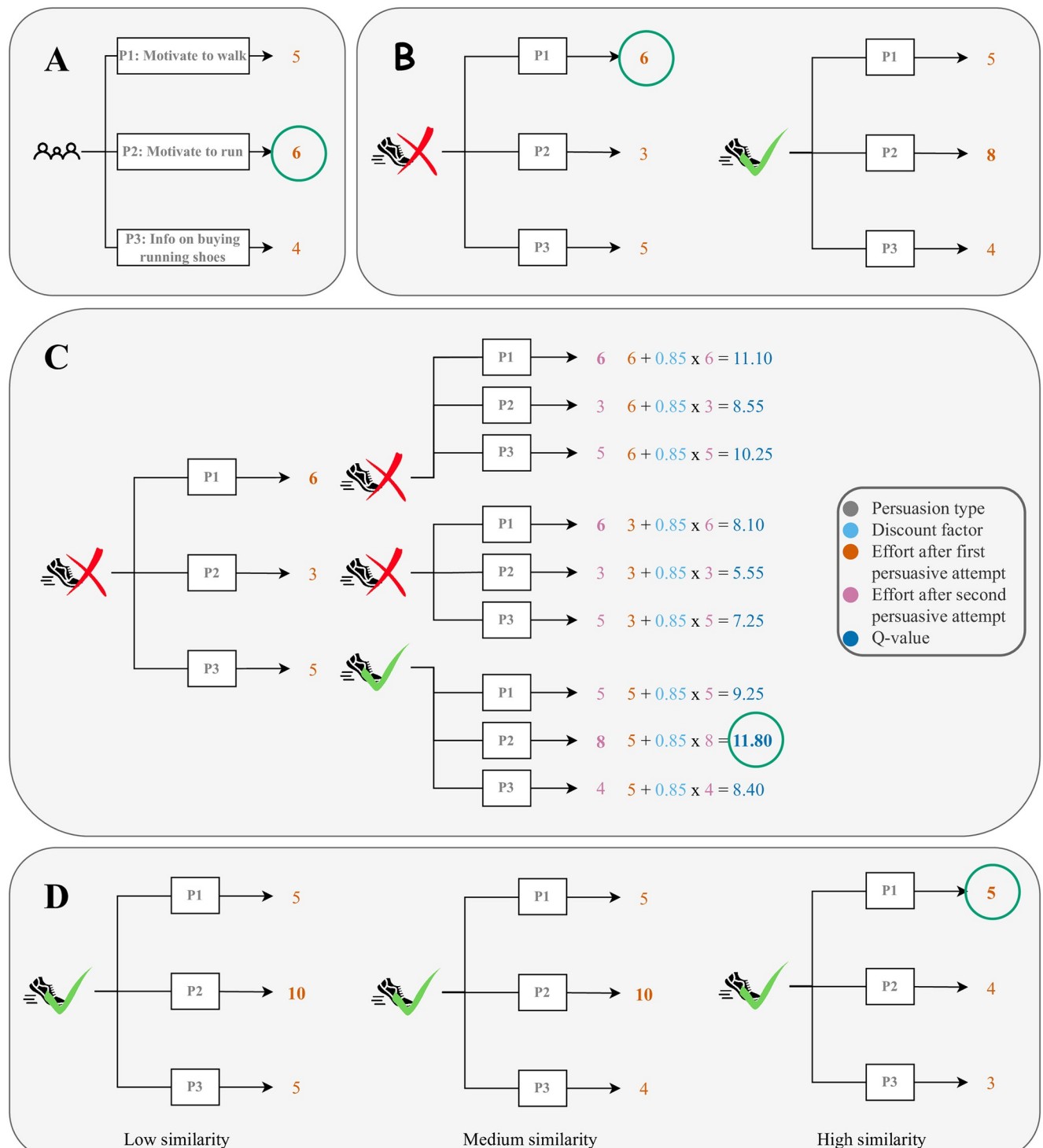

**Fig 2. Illustration of the algorithm components.** Illustration of our proposed algorithm components. To the baseline of sending the most effective persuasion type (A) we add the consideration of states (B), next states (C), and the similarity of people (D). Circles indicate the most effective persuasion type for the example person Martha described in the text.

choose persuasion type $P2$. However, intuitively, Martha's reaction to persuasive attempts might differ based on the state she is currently in. For instance, if she has no running shoes, just motivating her to go for a walk ($P1$) might be better than motivating her to go running (Fig 2B). Previous work has posited the importance of considering the context of a persuasive attempt when striving to create effective persuasion [25–27], for example, by defining the current state of the persuadee. This should be done so that knowing the persuadee's state allows one to predict the effectiveness of different persuasive messages. One such characteristic of a persuadee's state is the presence of barriers, such as Martha's lack of running shoes. Alfaifi et al. [40], for example, distinguish health, environmental, psychological, personal, and social barriers. Another potential state feature is self-efficacy, as it influences which health messages are more effective [28]. Moreover, how a person processes messages changes based on their mood [41, 42]. We, therefore, posit that choosing a message based on a persuadee's state is more effective than choosing the overall most effective message.

The effectiveness of persuasive attempts might depend on the persuadee's state, but a persuasive attempt in turn might also affect the state and thus the effectiveness of future persuasive attempts. Sending instructions on buying running shoes when Martha does not have any, for example, may cause Martha to buy some and thus remove the corresponding barrier. Future persuasive attempts that aim to increase Martha's motivation to run may then be more successful. Thus, even though informing Martha about buying running shoes with $P3$ may lead to less effort at the current time step than motivating her to go for a walk with $P1$, the former may allow the virtual coach to more successfully motivate Martha in the future (Fig 2C). To estimate the overall effectiveness of $P3$, we can compute the discounted sum or Q-value of the efforts after $P3$ at the current time step and the most effective persuasion type at the next time step. Discounting thereby means that we give a lower weight to efforts in the more distant future due to the importance of initial small wins [43]. In the example in Fig 2C, the discount factor is set to 0.85, and the discounted sum or Q-value is with 11.8 the largest if we choose $P3$ at the current step. Regarding the previously mentioned state features from the literature, Steward et al. [29] found that differently framed messages vary in their impact on self-efficacy. Given that self-efficacy determines how effective different health messages are [28], a message choice at this time point thus determines the effectiveness of messages in the future. Besides self-efficacy, the type of message might also affect a person's intention to act, anticipated regret, and attitude toward behavior [44]. We thus hypothesize that selecting a message based on both the present and the future states of a persuadee is altogether more effective than considering only the persuadee's present state and choosing the overall most effective message.

A person's state can change frequently, so we need to infer it each time we make a persuasive attempt. However, there are also relevant characteristics of a person which change, if at all, very slowly. For example, the impact of message types on self-efficacy depends on a person's need for cognition [29]. Other variables that may affect the success of different messages include the stage of behavior change [20], personality [20, 32–38], age and gender [34, 45], cultural background [46], how people approach pleasure and pain [47, 48], self-construal or the perceived relationship between the self and others [49], and in the context of quitting smoking the experience with previous quit attempts [29]. Thus, we suppose that people who are more similar concerning such characteristics are more likely to respond similarly to persuasive attempts. When deciding how to persuade somebody, we thus want to weigh the data observed from other people based on how similar they are to the person at hand. For example, we may find that for people like Martha, it is better to motivate them to go for a walk ($P1$) than to go for a run ($P2$) once they have running shoes (Fig 2D). We hence posit that considering a persuadee's similarity to other people besides their current and future states when choosing a

persuasive message is more effective than not taking the similarity to other people into account. Overall, we thus hypothesize the following:

*H1: Subsequently incorporating 1) states, 2) the consideration of future states, and 3) the weighting of samples based on the similarity of people into an algorithm that selects the best persuasive message type is more effective than not incorporating the respective element.*

## H2: Similarity of optimal persuasion strategies

Previous work on persuasion algorithms claims the need for considering the domain. For example, Alslaity and Tran [25] found that the impact of persuasion types varies between domains such as e-commerce and movie recommendations. Intuitively, it is possible to continuously split domains into sub-domains such as e-commerce for clothes and e-commerce for books. Nevertheless, this is not done by persuasion approaches such as the ones by Alslaity and Tran [25] and Kaptein et al. [17]. The underlying assumption is that there is a certain level of domain granularity at which one can meaningfully generalize from one persuasive attempt to another. We, therefore, assume that we can persuade people similarly for preparatory activities for quitting smoking and those for increasing physical activity, as they serve the same behavioral goal of quitting smoking. Thus, we hypothesize the best persuasion strategy (i.e., policy) to be similar if we use data collected on both types of activities compared to using data collected on only one type of activity, or more formally:

*H2: The optimal policy is similar when learned based on a combined data set of activities for smoking cessation and increasing physical activity, and when learned based on a data set of activities for smoking cessation and on a data set of activities for increasing physical activity separately.*

## Methods

To test our hypotheses stated above, we conducted a longitudinal experiment from 20 May 2021 until 30 June 2021. The Human Research Ethics Committee of Delft University of Technology granted ethical approval for the research (Letter of Approval number: 1523). Before the collection of data, the experiment was preregistered in the Open Science Framework (OSF) [50].

### Experimental design

The experiment consisted of a prescreening to determine the eligibility of participants, a pre-questionnaire, five sessions in which a virtual coach attempted to persuade participants to do a new preparatory activity for quitting smoking or increasing physical activity, and a post-questionnaire. Participants were persuaded with a random persuasion type in the first two sessions and a persuasion type chosen by a persuasion algorithm after that.

Fig 3 shows the experimental design of the study. It was set up as a double-blind mixed-design study with two within-subject factors and one between-subject factor. The within-subject factors were the session in which a persuasive attempt was made (4 levels: sessions 1–4) and algorithm activeness (2 levels: off/on for sessions 1–2/3–4). The between-subject factor was the algorithm complexity used to choose a persuasion type after session 2. This factor had four levels with successively more elaborate optimization strategies. Ordered by complexity, the algorithm levels look for the highest value of either: 1) the average reward, 2) the average reward in a person's state, 3) the Q-value in a person's state, or 4) the similarity-weighted Q-value in a person's state. This means that starting from sending a persuasion type with the highest average reward, we progressively added the consideration of states, future states, and

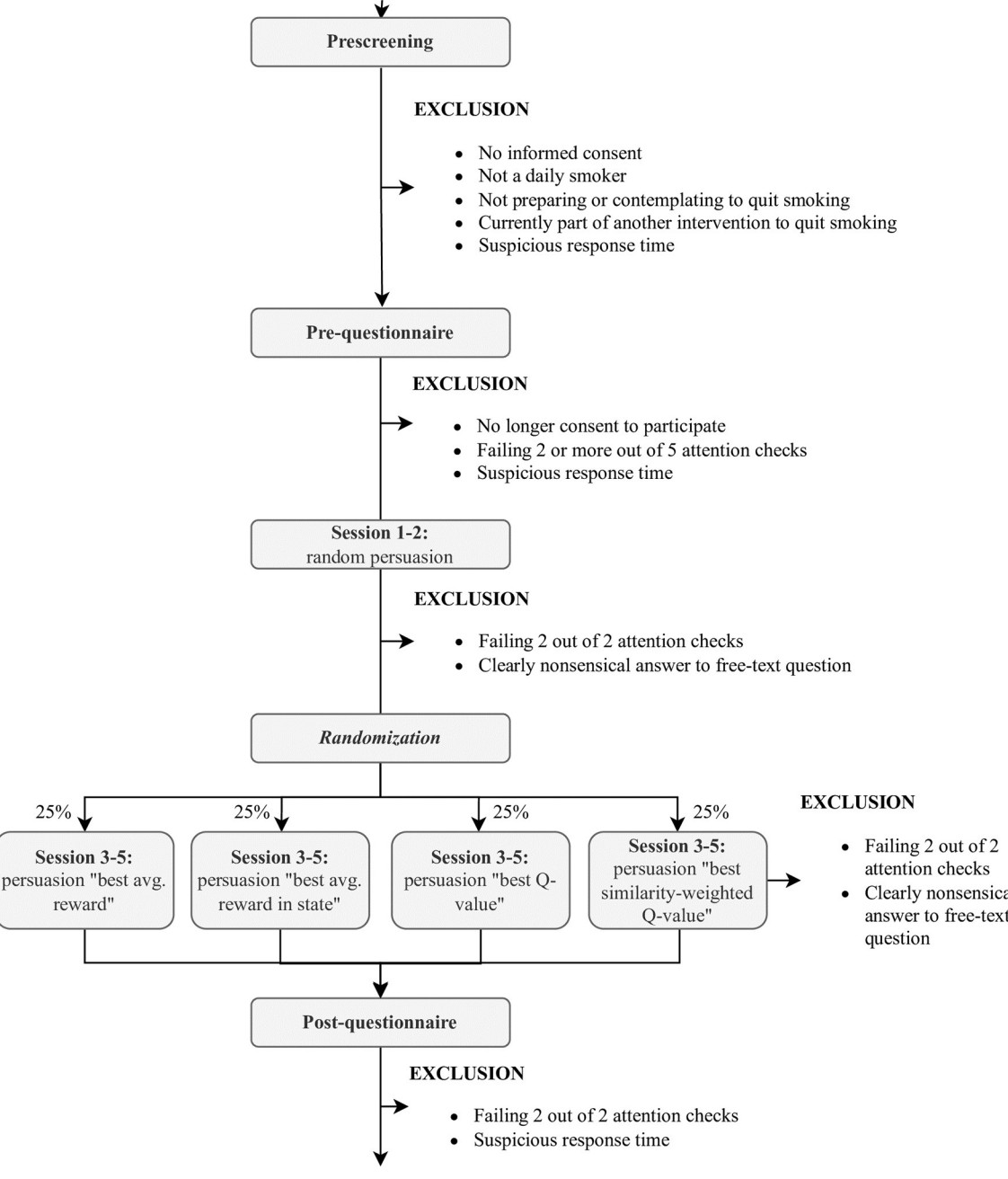

**Fig 3. Experimental design.** Design of the experiment, including the study components and in- and exclusion criteria for participants.

**Table 1. Chosen persuasion type for each algorithm complexity level.**

| Chosen Persuasion Type |
| --- |
| 1: BEST AVG. REWARD<br>The persuasion type $a$ with the overall highest average reward: $max_{a \in A}\{R(a)\}$. |
| 2: BEST AVG. REWARD IN STATE<br>The persuasion type $a$ with the highest average reward in a person's state $s$: $max_{a \in A}\{R(s, a)\}$. |
| 3: BEST Q-VALUE<br>The persuasion type $a$ with the highest Q-value in a person's state $s$: $max_{a \in A}\{Q^*(s, a)\}$. |
| 4: BEST SIMILARITY-WEIGHTED Q-VALUE<br>The persuasion type $a$ with the highest similarity-weighted Q-value in the state $s$ of person $i$: $max_{a \in A}\{Q_i^*(s, a)\}$. |

Abbreviations: $A$, Action space; $R(a)$, Average reward for taking action $a$; $R(s, a)$, Average reward for taking action $a$ in state $s$; $Q^*(s, a)$, Expected cumulative discounted reward for taking action $a$ in state $s$ and an optimal action in all subsequent states.

the weighting of samples based on the similarity of people. Table 1 provides an overview of the four complexity levels, whose components will be explained in the next section.

After session 2, we randomly assigned participants to one of the four algorithm complexity levels. Thereby, we aimed to balance the four groups regarding the potential covariates gender, Big-Five personality, stage of change for becoming physically active, and the effort participants put into their first activity. We used block randomization for the categorical gender variable, and adaptive covariate randomization for the other variables. Adaptive covariate randomization considers both previous assignments and covariates of people to balance condition assignments within covariate profiles [51]. Our approach to adaptive covariate randomization was a modification of the algorithm put forward by Xiao et al. [52].

## Algorithm

We created a virtual coach embedded in a conversational agent that attempted to persuade people to do small activities. For each persuasive attempt, the virtual coach selected a persuasion type based on its learned policy. After two to five days, the user provided the virtual coach with feedback by reporting the effort put into their activity. The virtual coach used this feedback to update its policy. Formally, we can define our approach as a Markov Decision Process (MDP) $\langle S, A, R, T, \gamma \rangle$. The action space $A$ thereby consisted of different persuasion types, the reward function $R: S \times A \times S \to [-1, 1]$ was determined by the self-reported effort, $T: S \times A \times S \to [0, 1]$ described the transition function, and the discount factor $\gamma$ was set to 0.85 to favor rewards obtained in the near future over rewards obtained in the more distant future. The intuition behind this value for $\gamma$ was that while we wanted to persuade a user over multiple time steps successfully, a failed persuasive attempt in the near future could cause a user to become less receptive to future ones or even to drop out entirely: early success might encourage people to continue [43]. The finite state space $S$ described the state a user was in and was captured by answers to questions about a user's capability, opportunity, and motivation to perform an activity [53]. The goal of an agent in an MDP is to learn an optimal policy $\pi^*: S \to \Pi$ $(A)$ that maximizes the expected cumulative discounted reward $\mathbb{E}[\sum_t^\infty \gamma^t r_t]$ for acting in the given environment. The expected cumulative discounted reward for taking action $a$ in state $s$ and an optimal action in all subsequent states is given by the Q-value function $Q^*: S \times A \to \mathbb{R}$. To incorporate the similarity of people, the virtual coach maintained a policy $\pi_i$ for each user $i$. When updating $\pi_i$, an observed sample from user $j$ was weighted based on how similar $i$ and $j$ were. We provide an overview of the algorithm component definitions in Table 2. In the following, we describe each algorithm component in detail.

**Table 2. Overview of the algorithm components and their definitions.**

| STATE SPACE |
|---|
| $S$ = {000, 001, 010, 011, 100, 101, 110, 111}, using three binary features based on the COM-B model (see the "State space"-section for more information). |

| ACTION SPACE |
|---|
| $A$ = {Commitment (Com.), Consensus (Con.), Authority (Au.), Action planning (AP), No persuasion (NP)} |

| REWARD |
|---|
| $$r = \begin{cases} -1 + \dfrac{e}{\bar{e}} & if \; e < \bar{e} \\ 1 - \dfrac{10 - e}{10 - \bar{e}} & if \; e > \bar{e} \\ 0 & otherwise, \end{cases}$$ |
| where $e \in [0, 10]$ is an effort response and $\bar{e}$ the mean effort. |

| REWARD FUNCTION |
|---|
| $R: S \times A \times S \rightarrow [-1, 1]$ such that $R(s, a, s')$ is the instant reward for taking action $a$ in state $s$ and arriving in state $s'$. |

| TRANSITION FUNCTION |
|---|
| $T: S \times A \times S \rightarrow [0, 1]$ such that $T(s, a, s') = Pr(s'|s, a)$ is the probability of arriving in state $s'$ after taking action $a$ in state $s$. |

| DISCOUNT FACTOR |
|---|
| 0.85 |

| SIMILARITY COMPUTATION |
|---|
| 1. Scale TTM-stage and five personality dimensions to the interval [0, 1]. |
| 2. Compute for a person $i$ her similarity to all other people $j$ based on the Euclidean distance between their six-dimensional trait vectors, whereby the largest distance is mapped to a similarity of 0 and the smallest distance to a similarity of 1. |
| 3. Compute the weight $w_{ij}$ of a sample from person $j$ for person $i$ as follows: |
| $$w_{ij} = max\left( \frac{s_{ij}}{\sum_k s_{ik}}, 0.0001 \right),$$ |
| where $s_{ij}$ is the similarity of $i$ and $j$. |

Abbreviations: COM-B model, Capability-Opportunity-Motivation-Behavior model; TTM, Transtheoretical model.

**State space.** Each session, participants answered ten questions on a 5-point Likert scale. Seven of these questions were based on the Capability-Opportunity-Motivation-Behavior (COM-B) self-evaluation questionnaire [53]. According to the COM-B model, capability, opportunity, and motivation together determine one's behavior, which in turn influences one's capability, opportunity, and motivation [54]. This made using capability, opportunity, and motivation as state variables for our RL approach appealing. We supplemented the seven questions from the self-evaluation questionnaire with people's self-efficacy due to the impact of self-efficacy on the effectiveness of different types of persuasive messages [28]. In addition, we asked people about their smoker and physical activity identities as according to Prime theory, self-identity can be a reliable predictor of behavior [55]. These additional questions fall under motivation in the COM-B model of behavior [56].

To lower the required amount of training data from the two training sessions, we subsequently reduced the size of the state space in two ways. First, we turned each state feature into a binary feature based on whether a value was greater than or equal to the feature mean (1) or less than the feature mean (0). Second, we selected three features in a way that was inspired by the G-algorithm [57]. Features were selected based on average rewards for level 2 and Q-values for levels 3 and 4 of algorithm complexity. The result of this state-space reduction was a state space of size $2^3 = 8$ (see Table 5 for the chosen features).

**Action space.** Five persuasion types defined the action space. These were authority, commitment and consensus from Cialdini [58], action planning [59], and no persuasion. For each

**Table 3. Reflective questions for authority, commitment, and consensus.**

| Persuasion Type | Reflective Question: "Please tell me what you think: . . . |
|---|---|
| Authority | Which other experts, whose opinion you value, would agree with this?" |
| Commitment | In what way does doing this activity match your decision to successfully quit smoking?" |
| Commitment–Identity | In what way does doing this activity match your decision to become somebody who has successfully quit smoking?" |
| Consensus | How would people like you, in a situation like yours, agree with this?" |

persuasion type, we formulated a set of message templates that were filled in for each activity to obtain persuasive messages. In the case of Cialdini's persuasion types, we created these templates by removing the domain-specific information from the validated healthy eating messages by Thomas et al. [36]. Due to the importance of self-identity in behavior [55], we also added two identity-based message templates for the commitment persuasion type. For action planning, we created templates based on the formulation by Sniehotta et al. [60]. However, rather than asking participants to enter their action plans in a table, the virtual coach prompted them to create an if-then plan of the form "If ⟨situation⟩, then I will ⟨do activity⟩" based on Chapman et al. [61]. In addition, the virtual coach provided an example of such an if-then plan as recommended by Hagger and Luszczynska [59]. S1 Appendix lists examples of the resulting templates and persuasive messages.

The virtual coach asked the participants to type their action plans into the chat, which indicated whether the participants had read the message. In the case of the three persuasion types from Cialdini, however, simply showing the persuasive messages may have meant that participants did not centrally process or even read the messages [62]. However, such central processing was desirable. As the elaboration likelihood model indicates, high-effort central processing of messages leads to attitudes that are more likely to be persistent over time, resistant to counterattack, and influential in guiding thought and behavior [63]. Therefore, we attempted to increase in-depth central processing of the persuasive messages in three ways. First, the virtual coach printed the persuasive messages in boldface [64] to reduced distraction [63]. Second, the virtual coach asked participants to answer reflective questions to increase self-referencing [63, 65] (Table 3). Third, we repeated the persuasion type [63] by adding reminder questions to the reminder messages participants received after each session (Table 4). We based these reminder questions on the ones used by Schwerdtfeger et al. [62] to remind people of their action plans and sent them for Cialdini's persuasion types as well as action planning.

**Reward.** In sessions 2–5, participants were asked about the overall effort they put into their previously assigned activity on a scale from 0 to 10. Based on the mean effort $\bar{e}$ computed after session 2, the reward $r \in [-1, 1]$ for an effort response $e$ was computed as follows:

$$r = \begin{cases} -1 + \dfrac{e}{\bar{e}} & if\ e < \bar{e} \\ 1 - \dfrac{10 - e}{10 - \bar{e}} & if\ e > \bar{e} \\ 0 & otherwise. \end{cases}$$

**Table 4. Examples of reminder question templates.** Examples of templates for the reminder questions that are added to the reminder messages people receive.

| Persuasion Type | Example of Reminder Question Template |
|---|---|
| Action planning | Keep in mind your rule for ⟨doing activity⟩ before the next session! |
| Authority | Do you remember which experts, whose opinion you value, would argue that ⟨doing activity⟩ may help to ⟨positive impact of activity⟩? |
| Commitment | Recall how ⟨doing activity⟩ is in line with your decision to successfully quit smoking! |
| Consensus | Don't forget how people like you, in a situation like yours, would testify that ⟨doing activity⟩ may help to ⟨positive impact of activity⟩! |

The idea behind this reward signal was that an effort response that was equal to the mean effort was awarded a reward of 0, and that rewards for efforts greater and lower than the mean were each equally spaced.

**Reward and transition functions.**   The reward and transition functions were estimated based on the samples collected from the first batch of people ($N = 516$) who successfully completed session 2. No updates to these training samples were made afterward as more data was collected.

**Similarity computation.**   Rather than choosing the same persuasion type for each person in a state, the virtual coach maintained a separate policy $\pi_i$ for each user $i$. When computing $\pi_i$, an observed sample from user $j$ was weighted based on how similar $i$ and $j$ were. The virtual coach computed the similarity based on people's Big-Five Personality [66] and Transtheoretical Model (TTM)-stage [67] for becoming physically active. We chose these variables due to extensive previous work showing their impact on the success of different forms of persuasion [20, 32–38]. We did not consider the TTM-stage for quitting smoking, as participants had to be in one of two specific stages (i.e., contemplation or preparation) to be eligible for the study. For the similarity computation, the virtual coach first scaled the TTM-stage and the five personality dimensions to the interval [0, 1] so that the features had the same scale. Next, the virtual coach computed for a person $i$ her similarity to all other people $j$ based on the Euclidean distance between their six-dimensional trait vectors. Thereby, the virtual coach mapped Euclidean distances to similarities so that the similarity for the smallest Euclidean distance was 1, and the similarity for the highest Euclidean distance was 0. Lastly, the virtual coach computed the weight $w_{ij}$ of a sample from person $j$ for person $i$ as follows:

$$w_{ij} = max\left(\frac{s_{ij}}{\sum_k s_{ik}}, 0.0001\right), \tag{1}$$

where $s_{ij}$ is the similarity of $i$ and $j$, $k$ denotes a person on which samples were gathered, and the addition of 0.0001 was to ensure that no sample was given a weight of 0.

These similarity-based sample weights affected how the reward and transition functions were estimated for a person. For example, given a training set with one sample of the form $\langle s_k, a_k, r_k, s'_k \rangle$ from each of $K$ people, the reward $R_i(s, a, s')$ for person $i$ was computed as so:

$$R_i(s, a, s') = \frac{\sum_{k \in K, s_k=s, a_k=a, s'_k=s'} w_{ik} r_k}{\sum_{k \in K, s_k=s, a_k=a, s'_k=s'} w_{ik}}. \tag{2}$$

## Algorithm training

The persuasion algorithm on all four complexity levels was trained based on the data gathered in sessions 1 and 2 for the first batch of people ($N = 516$) who successfully completed session 2 (see Table 5 for the resulting policies). No samples were later added to this training set of 516 samples so that the policies for all people were trained on the same number of samples and hence comparable. For algorithm complexity levels 3 and 4, Q-values were computed via value iteration based on the estimated reward and transition functions.

The number of samples used to train the algorithm complexity levels was based on the guidelines by Cohen [85] for multiple regression analysis and a medium effect, an alpha of 0.05, and three independent variables (i.e., the three state features that describe the state space). This resulted in a sample size of 76. Since we have five actions, we multiplied this sample size by five for a value of 380. We used with 516 more than 380 samples as we had more people in the first batch of people who successfully completed session 2. Moreover, we analyzed the impact of sample sizes ranging from 25 to 2300 on the Q-value estimation and

**Table 5. The learned policy used in sessions 3–5 for each algorithm complexity level.** The state feature selection and training of all policies were based on the data gathered in sessions 1 and 2 for the first batch of people ($N = 516$) who successfully completed session 2. This training set of 516 samples was not updated thereafter as more data was gathered.

| Learned Policy | | | | | | | | |
|---|---|---|---|---|---|---|---|---|

**1: BEST AVG. REWARD**
*Commitment*

**2: BEST AVG. REWARD IN STATE**

| State Feature | State | | | | | | | |
|---|---|---|---|---|---|---|---|---|
| | 1 | 2 | 3 | 4 | 5 | 6 | 7 | 8 |
| Feeling like wanting to do an activity (F5) | 0 | 0 | 0 | 0 | 1 | 1 | 1 | 1 |
| Feeling like being part of a group that is doing these kinds of activities (F4) | 0 | 0 | 1 | 1 | 0 | 0 | 1 | 1 |
| Thinking they can do an activity (F8) | 0 | 1 | 0 | 1 | 0 | 1 | 0 | 1 |
| *Action* | *AP* | *AP* | *Con.* | *NP* | *Con.* | *NP* | *NP* | *Com.* |

**3: BEST Q-VALUE**

| State Feature | State | | | | | | | |
|---|---|---|---|---|---|---|---|---|
| | 1 | 2 | 3 | 4 | 5 | 6 | 7 | 8 |
| Thinking that it would be a good thing to do an activity (F7) | 0 | 0 | 0 | 0 | 1 | 1 | 1 | 1 |
| Thinking they can do an activity (F8) | 0 | 0 | 1 | 1 | 0 | 0 | 1 | 1 |
| Knowing why it is important to do an activity (F1) | 0 | 1 | 0 | 1 | 0 | 1 | 0 | 1 |
| *Action* | *Con.* | *Con.* | *Com.* | *NP* | *Au.* | *Com.* | *Con.* | *Com.* |

**4: BEST SIMILARITY-WEIGHTED Q-VALUE**

| State Feature | State | | | | | | | |
|---|---|---|---|---|---|---|---|---|
| | 1 | 2 | 3 | 4 | 5 | 6 | 7 | 8 |
| Thinking that it would be a good thing to do an activity (F7) | 0 | 0 | 0 | 0 | 1 | 1 | 1 | 1 |
| Thinking they can do an activity (F8) | 0 | 0 | 1 | 1 | 0 | 0 | 1 | 1 |
| Knowing why it is important to do an activity (F1) | 0 | 1 | 0 | 1 | 0 | 1 | 0 | 1 |
| *Action* | *Con.* | *Con.* | *Com.* | *NP* | *NP* | *Au.* | *Con.* | *Com.* |

The policy above is an example for one person (there was a separate policy for each person).

Abbreviations: Avg., Average; F, Feature; *AP*, Action planning; *Con.*, Consensus; *NP*, No persuasion; *Com.*, Commitment; *Au.*, Authority.

optimality of chosen actions. Specifically, we estimated the reward function, transition function, and the resulting "true" Q-values and optimal policy based on all 2366 samples gathered in our study. We then randomly drew different numbers of samples from these 2366 samples and computed the mean $L_1$-error for predicting the true Q-values based on 100 repetitions per sample size. We obtained a mean $L_1$-error of 0.68 for our sample size of 516, which is a reduction by more than two thirds of the mean $L_1$-error for a sample size of 25. In addition, the mean $L_1$-error for the true Q-values of the estimated optimal actions compared to the true optimal actions per state is only 0.08. This shows that the optimal action chosen based on 516 samples is only slightly worse than the true optimal action. S9 Appendix provides further information on this.

## Materials

We used four online services in this study: Prolific for recruiting, inviting, and communicating with participants, Qualtrics for hosting the questionnaires and instructions for the conversational sessions, and Google Compute Engine to host the virtual coach and the sessions via Rasa X. In addition, some activities assigned in the sessions involved watching a video on YouTube.

**Virtual coach.**   The virtual coach used for the sessions was implemented in Rasa [68]. It had the name Sam, which may help to increase its social presence [69]. Sam presented itself as

being there to help people to prepare to quit smoking and become more physically active as the latter may aid the former. In its responses, Sam used techniques from motivational interviewing [70] such as giving compliments for putting much effort into assigned activities and expressing empathy otherwise. Empathy can thereby also help to form and maintain a relationship with a user [71], which can support behavior change [69]. Based on discussions with smoking cessation experts, Sam maintained a generally positive and encouraging attitude while trying to avoid responses that may be perceived as too enthusiastic [72]. To make the conversations accessible for people with low literacy levels, large chunks of text were broken up into multiple messages, in between which participants had to indicate when to proceed. In addition, participants communicated mainly by clicking on buttons with possible answer options. Only when free-text input was crucial, such as when writing about the experience with an assigned activity, were buttons not used. Lastly, to avoid repetitiveness, Sam randomly chose from several different formulations for an utterance. This is important, as repetitiveness can negatively influence the engagement with a system and motivation to perform an advocated behavior [71]. The implementation of the virtual coach can be found online [73].

**Preparatory activities.** In each session, the virtual coach asked participants to complete a new preparatory activity for next time that related to quitting smoking or increasing physical activity, such as writing down and ranking reasons for quitting smoking:

*Having high aspiration to quit smoking may aid quitting successfully. So, before the next session, I advise you to identify and write down reasons why you want to stop smoking. After writing them down, think about which reasons are most important to you and order them accordingly.*

The virtual coach selected the activities from a pool of 24 activities of similar duration, 12 each for quitting smoking and increasing physical activity. The activities for quitting smoking were based on components of the StopAdvisor smoking cessation intervention [74] and future-self exercises [75, 76]. The ones for increasing physical activity were generated by adapting the smoking cessation activities. Each activity formulation included reasoning for why the activity could help to prepare to quit smoking. A psychologist and smoking cessation expert read through the activity formulations to ensure they were suitable and clear. The virtual coach proposed one activity for quitting smoking and one for increasing physical activity in the first two and the subsequent two sessions to each participant. The virtual coach chose the type of activity in the fifth session randomly. It selected an activity for an activity type uniformly at random while avoiding repetitions of the same and very similar activities (e.g., creating a personal rule for not smoking and creating a personal rule for becoming more physically active). So participants were never asked to do an activity more than once, as the goal was not to create habits. The formulations of the activities are provided in S8 Appendix.

## Measures

**Primary measures.** To assess the effectiveness of subsequently adding the algorithm components for our first hypothesis, we used the following primary measures:

*Effort.* The virtual coach measured the effort by asking participants about the effort they put into their previously assigned activity on a scale from 0 ("Nothing") to 10 ("Extremely strong"). The scale was adapted from Hutchinson and Tenenbaum [77]. Note that this effort measure also served as a basis for choosing persuasion types in the four algorithm complexity levels.

*Perceived motivational impact.* The virtual coach measured the perceived motivational impact of the sessions by asking participants "Please rate the impact of our last 2 conversations

on your motivation to do your previous assigned activities" at the beginning of their third and fifth sessions. The virtual coach prompted participants to enter any number between -10 and 10, with -10 being "Very negative," 0 being "Neutral" and 10 being "Very positive."

**Secondary measures.** *Algorithm input measures.* We measured several variables as input for the persuasion algorithms. This included ten possible state features, seven of which were adapted from the COM-B self-evaluation questionnaire [53] (e.g., "I know why it is important to do the activity") and answered on a 5-point Likert scale. The other three features were based on measuring self-efficacy [60] on a 5-point Likert scale, smoker identity [78] with the additional answer option "Smoker," and physical activity identity. For the latter, we adapted the item with the highest factor loading from the Exercise Identity Scale [79] to physical activity and asked participants to rate it on a 5-point Likert scale. The highest algorithm complexity level further required computing how similar people were. We accomplished this by using people's Big-Five personality based on the 10-item questionnaire by Gosling et al. [66] and TTM-stage for becoming physically active based on an adaptation of the question by Norman et al. [80] to physical activity.

*Activity involvement.* For exploration purposes, we further measured participants' involvement in their assigned activities in the post-questionnaire. We, therefore, asked participants to rate three items to assess whether they found their assigned activities interesting, personally relevant, and involving. The three items were based on Maheswaran and Meyers-Levy [81] and answered on a scale from -5 ("Disagree strongly") to 5 ("Agree strongly").

*Potential covariates.* We collected data on potential covariates for the first hypothesis in the pre-questionnaire. Besides the variables discussed above, this included quitter self-identity measured with three items based on Meijer et al. [82], the need for cognition based on the three items from Cacioppo et al. [83] used by Steward et al. [29], and physical activity identity based on an adaptation of the Exercise Identity Scale by Anderson and Cychosz [79] to physical activity. All items were rated on 5-point Likert scales.

*Data for future research.* We measured several other variables for future research. These variables are not discussed in this paper but are described in our OSF preregistration form [50].

## Participants

Prior to the experiment, we computed a conservative estimate of the sample size required for evaluating the effectiveness of the four algorithm complexity levels using the Monte Carlo simulation described in Chapter 4.9.2 in Chechile [84]. We ran the simulation based on the ability to find an effect size (Cohen's $g$) of 0.1, which is halfway between a small ($g = .05$) and a medium ($g = .15$) effect size according to Cohen [85], with reliable Bayes factor values of 19 or more for four conditions and a binary response variable. The result was a sample size of 132 per algorithm complexity level, resulting in a total of 528 participants. This estimate was conservative, as we had interval dependent variables instead of binary ones. In reaching the sample size, we were constrained by a budget limit of 5,000 euros.

Moreover, we conducted a Bayesian power analysis based on a Monte Carlo approach. We used 500 simulations of two conditions with a medium difference of 0.3 [85] between their standard normally distributed means. For each simulation, we computed the Bayes factor for the hypothesis that the mean of the second condition is higher than the one of the first condition. The power was then calculated as the fraction of simulations in which the Bayes factor was at least 19. The result was a power of 0.78 for 129 samples per condition, which is the lowest number of samples we obtained for an algorithm complexity level for the last session.

To be eligible, participants had to be fluent in English, smoke tobacco products daily, contemplate or prepare to quit smoking [86], not be part of another intervention to quit smoking,

and provide informed consent. In addition, we used the quality measures on Prolific to choose people who had completed at least one previous study and an approval rate of at least 90% for their previously completed studies. 1406 participants started the prescreening questionnaire, and 521 of the 922 eligible participants successfully reported on all their assigned activities in sessions 2 to 5. Participants were not invited to a subsequent study component when doing an entire component twice or failing two or more attention checks. In addition, participants had to respond to a study component invitation within about one day for the pre-questionnaire, three days for the sessions, and seven days for the post-questionnaire. The participant flow is depicted in S2 Appendix. Participants who completed a study component were paid based on the minimum payment rules on Prolific, which require a payment rate of five pounds sterling per hour. Participants were informed that their payment was independent of how they reported on their suggested preparatory activities to account for self-interest and loss aversion biases [87]. Self-interest bias can arise when incentives exist that motivate participants to respond in a certain way; loss aversion bias can occur when participants choose to not partici-pate or to drop out when suspecting that they may not be paid fairly. Participants who failed two or more attention checks in a study component were not reimbursed.

S3 Appendix lists participant characteristics such as age, gender, TTM-stage for quitting smoking, and the existence of previous quit attempts for each algorithm complexity level. We compared Bayesian models with and without the algorithm complexity level as a predictor for each characteristic to test for systematic differences between the four levels. We found that based on the Widely Applicable Information Criterion (WAIC), the model with the algorithm complexity level as a predictor always performed worse than the model without this predictor, therefore providing no indication of systematic differences between the algorithm complexity levels for these characteristics. Participants were nationals of diverse countries such as the United Kingdom, Portugal, the Russian Federation, the United States, Chile, South Africa, Nigeria, Turkey, India, Malaysia, and Australia.

## Procedure

We opened the intake for the prescreening questionnaire on Prolific on each of seven different days between 20 May and 8 June 2021. Each time, we invited as many people as we could afford to reach but not exceed our budget limit in case of no further dropout. Participants meeting the qualification criteria could access the prescreening questionnaire on Prolific, and those who passed the prescreening were invited to the pre-questionnaire about one day later. In the pre-questionnaire, we collected participant data such as the Big-Five personality and TTM-stage for becoming physically active. One day after completing the pre-questionnaire, we invited participants to the first of five sessions in which the virtual coach Sam assigned them a preparatory activity together with a persuasion type. Participants received instructions on interacting with Sam in Qualtrics before being directed to a website for the conversation. The structure of the conversations is depicted in S4 Appendix and two excerpts of actual con-versations are shown in S5 Appendix. Each session lasted about five to eight minutes, and invi-tations to a subsequent session were sent about two days after having completed the previous one. The study ended with a post-questionnaire, to which participants were invited about two days after completing the last session.

## Data preparation and analysis strategies

First, we corrected entry errors from state and attention check questions in the sessions that participants messaged us about on Prolific ($N = 4$). The corrections for state questions ($N = 2$) pertained to the question on smoker identity, with two participants correcting their entries

from "non-smoker" and "ex-smoker" to "smoker" for session 1. As participants were persuaded randomly in session 1, these entry errors did not affect the conversations. Entry errors for attention check questions ($N = 2$) had no effect on the conversations irrespective of the session. Next, we preprocessed the gathered data by 1) using only data from sessions and the post-questionnaire if people passed at least one attention check during the respective component, and 2) using the first recorded submission for a study component if people did the component more than once. In the following, we describe our data and analysis strategies for each hypothesis in detail. All data and analysis code can be found online [88].

**H1: Algorithm effectiveness.** We conducted a multi-level (i.e., hierarchical) Bayesian analysis of the data.

*Further data preparation*. We removed the data of people who did not complete session 2 and were therefore not assigned to a condition ($N = 5$). To make an exploratory analysis of subgroups based on activity involvement possible, we computed the reliability of the corresponding three items. As the reliability was sufficiently high (Cronbach's $\alpha = 0.89$), we used the mean of the items as an index measure.

*Statistical models*. We created three statistical models. For both dependent variables, the effort people put into their activities and the perceived motivational impact of the sessions, we fit models that contained a general mean, a random intercept for each participant, a fixed effect for algorithm activeness, a fixed effect for the algorithm complexity level, and a fixed interaction effect between the algorithm complexity level and algorithm activeness. For the effort, we also fit a second model that additionally included a main fixed effect for the session, as well as fixed interaction effects between the session and the other two factors. We fit all three models with diffuse priors based on the ones used by McElreath [89]. In addition, we performed a prior sensitivity analysis to assess the impact of using different priors. We found only a limited effect on the posterior probability for a hypothesis being true, as it changed by at most 0.02. A $t$-distribution was fit for the dependent variable in each model.

*Covariates*. We explored potential covariates such as the type of the assigned activities (i.e., quitting smoking or increasing physical activity), physical activity identity, and quitter self-identity. Adding these variables did not change the conclusions drawn about our hypothesis, and therefore we did not include the variables in the final models.

*Inference criteria*. For each of the three statistical models, we computed the posterior probability that our hypothesis was true based on samples drawn from the estimated model. This means that we evaluated for each model the posterior probability that the relevant parameter was greater than 0. For the first two models, this parameter was the fixed two-way interaction effect between the algorithm complexity level and algorithm activeness. For the third model it was the fixed three-way interaction effect between the algorithm complexity level, algorithm activeness, and the session. We interpreted posterior probabilities using the guidelines by Chechile [84] and their extension to values below 50% based on Andraszewicz et al. [90]. We also report the 95% Highest Density Intervals (HDIs) for the parameters, with an HDI being "the narrowest interval containing the specified probability mass" [89]. In addition, we used the Region of Practical Equivalence (ROPE) method [91] as a secondary method to evaluate the results. This method allows one to accept or reject a hypothesis or to withhold a decision. As the results of this method were all inconclusive, we do not report them.

*Implementation*. All analyses were carried out in R with the rethinking package [89]. We provide code to reproduce the analyses in a Docker container as recommended by van de Schoot et al. [92].

*Exploratory subgroup analysis based on activity involvement*. We divided participants into subgroups based on whether their activity involvement was greater than or equal to the median ($N = 269$) or less ($N = 231$). The analyses for the three models were then repeated separately

for each subgroup. Note that since we measured the activity involvement in the post-questionnaire, this analysis only included participants for whom we had data on at least one passed attention check from the post-questionnaire ($N = 500$).

**H2: Similarity of optimal persuasion strategies.** *Data preparation*. We compared the optimal policies computed based on all collected data to using only the data on activities for either quitting smoking or increasing physical activity. Therefore, we distributed the gathered data over three datasets based on the activity type. This resulted in 1175 samples for quitting smoking, 1191 for increasing physical activity, and 2366 for both activity types together.

*Analysis plan*. We computed the optimal policies for each non-baseline algorithm complexity level (i.e., levels 2–4) for each dataset. To use equal amounts of data for both activity types, we randomly drew 1,000 samples from each activity type. This means that we used 1,000 and 2,000 samples, respectively, when computing a policy based on a single activity type and both activity types together. To account for the impact of this random selection, the sampling and subsequent optimal policy computation were conducted 100 times. Afterward, we concatenated the optimal policies for the 100 repetitions into a single list for each data type. In the case of multiple best actions for a state, one of the best actions was chosen uniformly at random.

*Inference criteria*. For each non-baseline algorithm complexity level, we calculated Cohen's $\kappa$ between the list of optimal policies based on activities for both quitting smoking and increasing physical activity, and the list of optimal policies computed based on only samples in which participants were advised to do a preparatory activity for either quitting smoking or increasing physical activity. The outcomes were interpreted based on the guidelines by Landis and Koch [93] shown in S6 Appendix. We also determined Cohen's $\kappa$ between the optimal policies computed based on different samples drawn from the same data type for exploratory purposes. This allowed us to draw conclusions about the consistency of policies computed on a certain data type.

*Implementation*. We provide code to reproduce the analyses in Python.

## Results

Tables 6 and 7 provide overviews of the mean effort and perceived motivational impact per algorithm complexity level and measurement moment. As the reward was computed based on the effort, the former gives an indication of the reward obtained by the virtual coach for the four algorithm complexity levels. Furthermore, S7 Appendix shows the mean effort per activity and activity type. To give some intuition for the size of the observed differences in Tables 6 and 7, we can divide the largest change between the first and last measurement moment for a complexity level by the standard deviation. The resulting effect size is 0.14 based on an

**Table 6. Mean effort per algorithm complexity level and measurement moment.**

| Complexity Level | Measurement Moment | | | |
|---|---|---|---|---|
| | **1** | **2** | **3** | **4** |
| 1: Best avg. reward | 5.55 (2.52) | 5.49 (2.74) | 5.49 (2.94) | 5.59 (2.88) |
| 2: Best avg. reward in state | 5.54 (2.54) | 5.20 (2.84) | 5.20 (3.09) | 5.46 (3.01) |
| 3: Best Q-value | 5.48 (2.56) | 5.40 (2.94) | 5.32 (2.99) | 5.88 (2.86) |
| 4: Best similarity-weighted Q-value | 5.45 (2.47) | 5.13 (2.81) | 5.29 (3.00) | 5.11 (3.12) |

Standard deviations are provided in parentheses.

Effort was measured on a scale from 0 to 10.

Abbreviations: avg., average.

**Table 7. Mean perceived motivational impact per algorithm complexity level and measurement moment.**

| | Measurement Moment | |
| --- | --- | --- |
| **Complexity Level** | **1** | **2** |
| 1: Best avg. reward | 5.01 (3.42) | 5.20 (3.48) |
| 2: Best avg. reward in state | 4.83 (3.61) | 5.23 (3.97) |
| 3: Best Q-value | 4.75 (3.22) | 5.40 (3.22) |
| 4: Best similarity-weighted Q-value | 4.64 (3.41) | 5.05 (3.43) |

Standard deviations are provided in parentheses.

Perceived motivational impact was measured on a scale from -10 to 10.

Abbreviations: avg., average.

observed change of at most 0.40 for the effort, and 0.20 based on an observed change of at most 0.66 for the perceived motivational impact. Both effect sizes are at most small according to Cohen [85].

## H1: Algorithm effectiveness

Between the two baseline sessions and the two sessions in which the algorithms were active, the largest increase in effort was observed in complexity levels 1 and 3 (Fig 4A). Quantifying these observations based on our Bayesian analysis, Table 8 reveals that it is not worth betting against higher complexity levels leading to a larger increase in effort, with the mean of the credible values showing a decrease of 0.05 in effort between complexity levels 1 and 4 when the algorithms are active. The HDI thereby ranges from -0.43 to 0.33, with only 39% and thus less than half of the credibility mass favoring higher complexity levels leading to a larger increase in effort. However, a detailed examination of Fig 4A suggests that there are differences between the two active sessions, which are sessions 3 and 4. Specifically, complexity level 3 exhibits a change from an effort similar or lower compared to level 1 in session 3 to the highest effort in session 4. Complexity level 4, on the other hand, shows a decrease in effort between the two active sessions.

These observations are found back when the fit model is extended with the session as a predictor and specifically a three-way interaction effect between algorithm complexity, algorithm activeness, and session. This fit model assigns a posterior probability of 0.70 to the hypothesis that the increase in effort between the two active sessions is larger for higher complexity levels

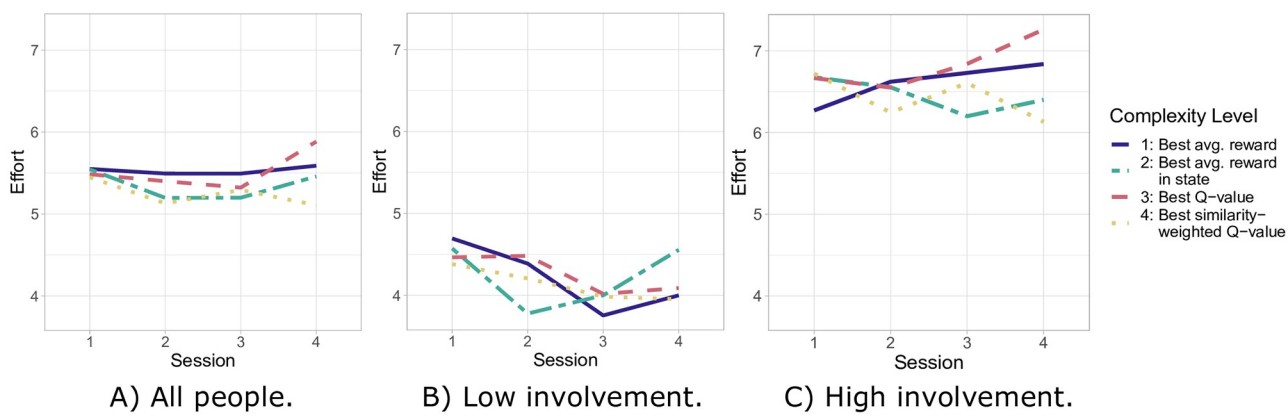

**Fig 4. Mean effort per session and algorithm complexity level.**

**Table 8. Results of Bayesian analyses of effort and perceived motivational impact.**

| DV | Parameter | Mean [HDI] (SD) | Post | Evaluation |
|---|---|---|---|---|
| Effort | alg. level × alg. activeness | -0.05 [-0.43, 0.33] (0.19) | 0.39 | Not worth betting against |
| | alg. level × alg. activeness × session | 0.20 [-0.57, 0.97] (0.40) | 0.70 | Not worth betting on |
| PMI | alg. level × alg. activeness | 0.09 [-0.34, 0.54] (0.22) | 0.67 | Not worth betting on |

Abbreviations: DV, Dependent variable; HDI, Highest density interval; SD, Standard deviation; Post, Posterior probability that a parameter's value is greater than 0; PMI, Perceived motivational impact.

(Table 8). As a result, more than half of the credibility mass are in favor of this hypothesis. Similarly, we observe in Fig 5A that complexity level 3 is accompanied by the largest increase in perceived motivational impact. Despite the increase being lower for complexity level 4 than for level 3, there is hence additional support for the first hypothesis. A Bayesian analysis confirms this (Table 8). More precisely, the posterior probability that the increase in perceived motivation impact is larger for higher complexity levels is 0.67. In other words, more than half of the credibility mass support this.

We also conducted separate analyses for people with high and low involvement in their activities for exploratory purposes. The mean effort is higher for people with high involvement (Fig 4C) than for those with low involvement (Fig 4B). For the high involvement subgroup, complexity level 1 shows the largest increase in effort between the two baseline and the two active sessions (Fig 4C). Quantitatively, the mean credible value is -0.20 with a posterior probability of only 0.20 that this increase is larger for higher complexity levels (Table 9). Again, however, complexity level 3 is associated with the largest increase in effort between the two active sessions (Fig 4C). This matches the quantitative results, according to which the posterior probability in favor of higher complexity levels leading to a larger increase in effort between the two active sessions is 0.96, a "good bet—too good to disregard" (Table 9). For people with low involvement, on the other hand, complexity level 1 is associated with the largest drop in effort between the two baseline and the two active sessions (Fig 4B). While this quantitatively leads to a posterior probability of 0.67 for higher complexity levels leading to a larger increase in effort (Table 9), none of the algorithm levels is very effective for this subgroup. Regarding the increase in effort between the two active sessions, complexity level 2 performs best but is still hardly effective in session 4 (Fig 4B). Therefore, based on our Bayesian analyses, it is not worth betting against higher complexity levels leading to a larger increase in effort between the two active sessions for this subgroup due to a posterior probability of 0.26.

Next, we conducted an exploratory analysis of the perceived motivational impact in the two subgroups. For people with high involvement (Fig 5C), the perceived motivational impact is much higher than for people with low involvement (Fig 5B). Thereby, complexity levels 3 and 4 show a larger increase in perceived motivational impact than levels 1 and 2 for the high involvement subgroup (Fig 5C). Given that this is the case only for complexity level 3 when both subgroups together are considered (Fig 5A), there seems to be more support for higher complexity levels leading to a larger increase in perceived motivational impact for people with high involvement. Quantitatively, the posterior probability in favor of this is 0.87, which can be qualified as a casual bet (Table 9). However, in contrast to the high involvement subgroup, we do not find much support for this for the low involvement subgroup. Neither of the four complexity levels is associated with an apparent increase in perceived motivational impact and instead levels 1 and 4 suggest even a slight decrease (Fig 5B). The posterior probability of 0.16 confirms this. In other words, it qualifies as a casual bet against higher complexity levels leading to a larger increase in perceived motivational impact for this subgroup (Table 9).

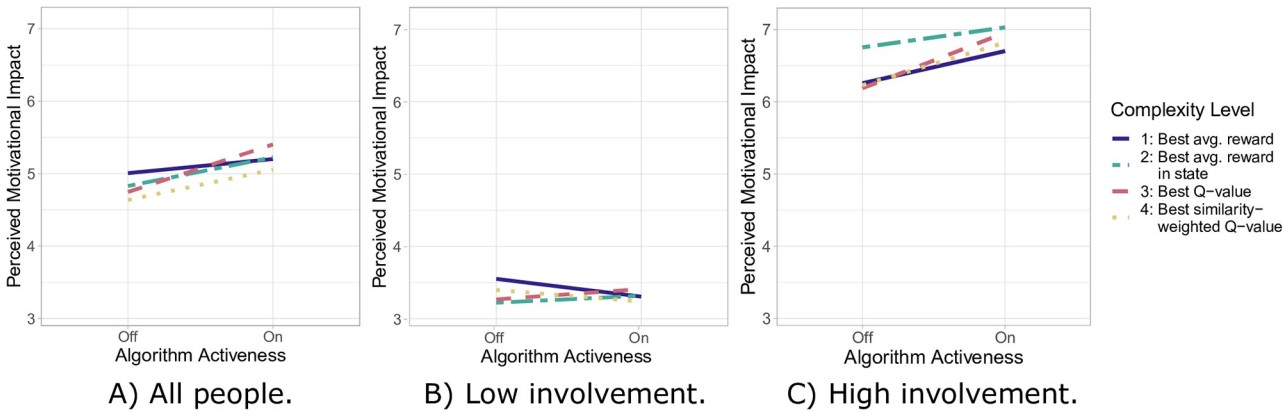

**Fig 5. Mean perceived motivational impact when the algorithms are off/on per algorithm complexity level.**

## H2: Similarity of optimal persuasion strategies

Table 10 shows fair to moderate agreement between the policies computed based on both activity types together on the one hand and activities solely for quitting smoking or increasing physical activity separately on the other hand for the non-baseline algorithm complexity levels. The agreement thereby tends to be much stronger for algorithm complexity level 3 and for physical activity also for algorithm complexity level 4. For reference, an upper limit of agreement was calculated by examining the agreement between policies computed from samples drawn from the same data set. This upper limit is moderate agreement for all three complexity levels.

## Discussion and conclusion

The presented longitudinal study examined the effectiveness of subsequently adding the consideration of states, future states, and the similarity of people to a personalized RL algorithm for persuading people to do preparatory activities for quitting smoking and increasing physical activity. The findings provide some support that people's reported motivation is positively affected by using higher algorithm complexity levels. The effort people spent on the activities also provides some support. Here, however, the overall advantage of using more algorithm elements becomes apparent only after some time, and initially, there seems to be no positive

**Table 9. Results of Bayesian analyses of effort and perceived motivational impact for people with low and high activity involvement.**

| DV | Parameter | Mean [HDI] (SD) | Post | Evaluation |
|---|---|---|---|---|
| LOW ACTIVITY INVOLVEMENT | | | | |
| Effort | alg. level × alg. activeness | 0.18 [-0.55, 0.94] (0.39) | 0.67 | Not worth betting on |
| | alg. level × alg. activeness × session | -0.45 [-1.83, 0.94] (0.71) | 0.26 | Not worth betting against |
| PMI | alg. level × alg. activeness | -0.45 [-1.36, 0.44] (0.46) | 0.16 | Only a casual bet against |
| HIGH ACTIVITY INVOLVEMENT | | | | |
| Effort | alg. level × alg. activeness | -0.20 [-0.67, 0.26] (0.24) | 0.20 | Only a casual bet against |
| | alg. level × alg. activeness × session | 0.80 [-0.10, 1.69] (0.46) | 0.96 | Good bet—too good to disregard |
| PMI | alg. level × alg. activeness | 0.26 [-0.20, 0.71] (0.23) | 0.87 | Only a casual bet |

Abbreviations: DV, Dependent variable; HDI, Highest density interval; SD, Standard deviation; Post, Posterior probability that a parameter's value is greater than 0; PMI, Perceived motivational impact.

**Table 10. Cohen's κ for algorithm complexity levels 2 to 4 after computing the optimal policies based on different types of data.** The types of data result from splitting the data based on the activity type, or using the data on both activity types together.

| Data Type | | Complexity Level | | |
|---|---|---|---|---|
| **1** | **2** | **2** | **3** | **4** |
| MAIN ANALYSES | | | | |
| Smoking cessation | Both | 0.33 | 0.36 | 0.26 |
| Physical activity increase | Both | 0.29 | 0.49 | 0.48 |
| REFERENCE ANALYSES | | | | |
| Both | Both | 0.60 | 0.54 | 0.58 |
| Smoking cessation | Smoking cessation | 0.54 | 0.56 | 0.42 |
| Physical activity increase | Physical activity increase | 0.51 | 0.57 | 0.59 |

impact. This is reflected by the three-way interaction effect in Table 8 and also visible in the increase in effort between session 3 and 4 for complexity level 3 in Fig 4A. Looking at the algorithm complexity levels separately, the level that considers current and future states by choosing a persuasion type with the highest Q-value seems most successful in moving people to future states in which they can be persuaded better. Support for this is even stronger for people who found the suggested activities most useful. An explanation may be that the persuasive messages have a stronger and more persistent impact on people with high activity involvement. According to the elaboration likelihood model, high involvement in an issue makes it more likely that messages are processed in detail [81]. Such in-depth processing in turn is more likely to have a persistent impact [63].

Extending the algorithm by weighting observed data samples based on the similarity of people did not perform well in this study. Specifically, the results suggest that the fourth algorithm complexity level that additionally considers the similarity of people based on their TTM-stage for becoming physically active and Big-Five personality is associated with a lower effort spent on the activities than the third level. This shows that increased personalization can be harmful, even if it is informed by literature, as in our case. One reason could be the necessity of using more domain-specific similarity variables such as quitter self-identity [82]. Future work can use our published data to determine whether such similarity variables are relevant in our domain. Moreover, while we computed similarity based on the Euclidean distances between vectors of user characteristics, other distances such as the cosine distance could be used (see Ontañón [94] for an overview).

Another interesting observation is that the impact of using higher algorithm complexity levels for people with low activity involvement appears to be not zero but in fact negative for the increase in effort between the two active sessions and the perceived motivational impact. This suggests that choosing persuasion types based on higher algorithm complexity levels is worse for this subgroup than doing so based on lower ones. The reason might be a novelty effect [95, 96]. A novelty effect arises because people are initially curious about a new system or technology and have high expectations. However, this curiosity and perceived usefulness fade over time as people become aware of the system's limitations. Applying this novelty effect to our study, participants likely had high expectations about the system's capability to help them prepare to quit smoking at the beginning. Afterward, the perceived usefulness of the approach may have decreased for some people as their expectations were not met. However, since we used the data gathered in the first two sessions as training data for the persuasion algorithms, the algorithms were trained mainly based on people who thought the system was useful. This likely lowered the performance of our algorithms overall, but especially the

performance of higher algorithm complexity levels for people with low activity involvement. This is the case because these higher algorithm complexity levels were fit more tightly to the data gathered from people with high activity involvement in the first two sessions. An important implication for future work is that it may be relevant to consider when a data sample was gathered during the behavior change process. Especially when persuasive attempts are made over a long time, might it be beneficial to give a lower weight to samples collected at the beginning of the interaction with the system. Furthermore, since people's preferences can also change over time [97], the weights for later samples need to be chosen carefully as well.

As to our second hypothesis, we see some agreement between the persuasion types chosen based on all collected data and the ones chosen based on data for only quitting smoking or increasing physical activity. This lends some support to transferring knowledge between these two activities types. Yet, we find that the agreement tends to be higher between optimal policies computed based on different samples of the same data type. This suggests that the algorithm could be improved by considering the activity type in the optimal policy computation. One reason for the lower agreement between optimal policies computed based on different data types might be that the involvement in the two activity types differed. We observed, for example, that people put overall less effort into activities for physical activity increase than ones for smoking cessation (S7 Appendix). Since the effort in our study was lower for people with low involvement in the activities, the processing of messages for the two activity types may have been different. Potentially, the link between becoming more physically active and quitting smoking could be made more evident. Besides higher agreement between optimal policies computed on the same data type, we also find that higher agreement between optimal policies computed based on different data types is achieved for algorithm complexity level 3 and for physical activity also for level 4. Thus, incorporating future states in algorithm complexity levels 3 and 4 and using people's TTM-stage for becoming physically active to weigh the observed samples in algorithm complexity level 4 appears to have helped capture the difference between the two activity types. It would be interesting to see in future work if other state or similarity variables could further improve upon this.

Besides the ideas mentioned above, there are many further directions for extending our work. First, it is interesting to think about the choice of reward signal. We want that people do their preparatory activities more thoroughly, so that they are better prepared for quitting smoking, so that they better achieve and maintain abstinence from smoking. While the more distal outcome measures in this chain capture the actual behavior we want to see, using them as reward signal to compare the four algorithm complexity levels leads to several challenges. This includes the time until we receive the signal (e.g., it may take several months or years before we know whether somebody has maintained their abstinence) and the signal's noisiness (e.g., a cancer diagnosis a year from now can also affect abstinence maintenance). In the case of being prepared for quitting smoking, an additional challenge is how this "preparedness" can be measured given that the activities differ in what they are meant to achieve, be it increasing self-efficacy or removing smoking cues. Certainly, several questions could be asked, but the number of questions should be kept low in light of the already low adherence rates for eHealth applications. Since the links in the chain from thoroughly doing preparatory activities to successful maintenance of smoking abstinence have already been supported by other literature (e.g., [74]), we thus chose the effort people spent on their activities as a more proximal reward signal. Notably, however, an even more proximal reward signal could be added. This is because our results suggest that with motivation, one of the predictors of behavior is increased to a greater extent if future states are taken into account. A combination of effort and perceived motivational impact could thus be used. Hiraoka et al. [98], for instance, use a reward signal that combines user satisfaction, the success of persuasion, and the naturalness of system

utterances. It may also be worthwhile to add a more objective measures of behavior than self-reported ones such as the effort in our study. This may, however, not be feasible for some (parts of) activities, such as placing a rule for not smoking in a place one can see every day.

Another direction for future work is to use our learned policy as a starting point and subsequently adapt to single individuals by focusing on their personal samples (rather than samples from similar people as in our fourth complexity level). The reason is that research has shown that the way people respond to a persuasive attempt is a good predictor of how they will respond to the same attempt in the future [17, 22–24]. Moreover, it may be important to consider the impact repeated actions can have. Repeatedly sending the same persuasion type may make it less effective [36], but could also help to strengthen the link between cue and response for action planning [62] or to scrutinize arguments objectively [99]. One interesting study in this regard is the one by Mintz et al. [22], which considers the effects both of repeating a message and of subsequently not sending the message for some time. Lastly, another avenue for future work is to ensure that the algorithms are ethical. For instance, it may not be ethical to choose a persuasion type that is predicted to be effective while at the same time lowering a person's self-efficacy. One way to incorporate such values or norms may be to learn a separate constrained policy from ethical examples [100] or expert preferences [101]. Other relevant issues are user trust, user privacy, and low bias [69].

On a higher level, our results show us that the impact of message-based persuasion algorithms on predictors of behavior and behavior itself is small. For example, the mean perceived motivational impact when the algorithms are active does not differ by more than 0.35 (Cohen's $d = 0.10$) between the four algorithm complexity levels (Fig 5A). This qualifies as less than a small effect size according to Cohen [85]. Similarly, the mean effort for session 3 does not differ by more than 0.29 (Cohen's $d = 0.10$) between the four algorithm complexity levels (Fig 4A). While other persuasive messages could have a larger effect, our findings are in line with other work. Kaptein et al. [17], for instance, found that the difference between a random and a tailored persuasive message with regards to the number of daily consumed snacks is 0.08 for a single persuasive attempt. Similarly, de Vries [20] saw that self-reported physical activity increases over time for both a tailored and a random message condition, with the physical activity being slightly but not significantly higher for the tailored condition. While the results of de Vries [20] and the snacking study of Kaptein et al. [17] were based on relatively small sample sizes of overall 47 and 73 participants, respectively, our results now show that the impact of persuasion algorithms on behavior is small even when conducting a large-scale experiment with at least 129 participants per condition and a resulting power of at least 0.78. Arguably, the impact of persuasion algorithms has been found to increase over time in both our experiment and the snacking study of Kaptein et al. [17]. More research is needed to test whether and how this increase in effectiveness occurs when persuasive attempts are made over more extended periods such as weeks or months. Even though the dangers of message amplification have been pointed out in various contexts such as social media [16], it is not yet well understood how and to which extent it influences actual behavior.

Given the so far limited impact of persuasive messages, an alternative may be to strategically persuade people through an entire dialog (e.g., [98, 102]). In addition, one could aim to further increase the processing of the persuasion by using multi-modal forms of persuasion (e.g., [97, 103, 104]). This choice of modalities can also be learned [105]. Alternatively, one could optimize the suggested behavior rather than the persuasive message. Our results show, for example, that the mean effort for the most effective complexity level in session 3 is by 2.84 (Cohen's $d = 0.95$) higher for people with high than for people with low involvement in their activities (Fig 4). In contrast to the difference between complexity levels, this qualifies as a large effect size [85]. Relevant approaches in this regard are ones to optimize suggested step goals [106,

107], activities for elderly people [108], physical [109] or learning [110] activities, or breast cancer screening recommendations [111].

In conclusion, we have presented a personalized RL-based persuasion algorithm and systematically tested the effectiveness of the algorithm components. Our results support the importance of taking states and future states into account to persuade people again. We expect that future work can build on these results to improve persuasion algorithms further. We make the dataset on 2366 persuasive messages sent to 671 people publicly available to facilitate this. Given the sparsity of public datasets in this field and the expensive nature of collecting data on human behavior, we think this helps those wishing to develop new algorithms or to test existing ones. For the field of behavior change, our dataset provides the effectiveness of different activities based on the effort people spent on them. This shows, for example, that the link between increasing physical activity and quitting smoking needs to be made more evident for participants. In addition, our results lend support to the COM-B model of behavior change, as state variables derived from this model showed to help predict behavior. Thus, our study can be seen as a successful example of combining computer science and behavior change theories to test behavior change theories in a large-scale experiment [112]. Given the fruitful insights for both fields, we encourage further work at their intersection.

## Supporting information

**S1 Appendix. Persuasive message examples.** Table that displays examples of persuasive message templates and resulting messages.
(PDF)

**S2 Appendix. Participant flow.** Figure that shows the participant flow through the study components in the experiment. The numbers next to the downward arrows denote how many people started the study components. We show the distribution across the four algorithm complexity levels for the participants who did not respond to the invitation to a study component after the randomization. Note that participants can return their submission on Prolific to withdraw from a study.
(PDF)

**S3 Appendix. Participant characteristics.** Table that depicts the participant characteristics for each algorithm complexity level.
(PDF)

**S4 Appendix. Conversation structure.** Figure that presents the structure of the five conversational sessions with the virtual coach Sam.
(PDF)

**S5 Appendix. Excerpts of actual conversations.** Figure that shows two excerpts of actual conversations with the virtual coach. The excerpts include the last state question and persuasion based on the persuasion types authority and action planning.
(PDF)

**S6 Appendix. Cohen's $\kappa$ interpretation guidelines.** Table that shows the interpretation guidelines for Cohen's $\kappa$ based on Landis and Koch [93].
(PDF)

**S7 Appendix. Mean effort per activity and activity type.** Table that shows for each activity the number of times participants reported the effort they spent on the activity and the mean effort that was reported. We include only the samples used in our analysis for H1 (i.e., not the

five samples of participants that were never assigned to a condition due to not completing session 2 and which were only used in the analysis for H2).
(PDF)

**S8 Appendix. Activity formulations.** Document that provides the formulations for the 24 preparatory activities that were used in the study. In case an activity involved watching a video, there were two different formulations for the session and the reminder message, with only the one for the latter containing the link to the video. This was to prevent participants from directly clicking on the video link when reading their activity during the session.
(PDF)

**S9 Appendix. Impact of sample size on Q-value estimation and optimality of chosen actions.** Figure that shows the mean $L_1$-errors based on 100 repetitions when drawing different numbers of samples from the 2366 samples we gathered. We provide mean $L_1$-errors for comparing the estimated and true Q-values for all state-action combinations (yellow) and comparing the true Q-values of the estimated and true optimal actions for all states (blue). True Q-values and optimal actions are those that are computed based on all 2366 samples. The horizontal lines indicate percentages of the mean $L_1$-error for the lowest number of samples compared to the highest number of samples for comparing the estimated and true Q-values for all state-action combinations.
(PDF)

## Acknowledgments

The authors would like to thank Dr. Eline Meijer for her feedback on the preparatory activities used in the study, and Mitchell Kesteloo for his help in running the virtual coach on a server.

## Author Contributions

**Conceptualization:** Nele Albers, Mark A. Neerincx, Willem-Paul Brinkman.

**Data curation:** Nele Albers.

**Formal analysis:** Nele Albers.

**Funding acquisition:** Mark A. Neerincx, Willem-Paul Brinkman.

**Investigation:** Nele Albers.

**Methodology:** Nele Albers, Willem-Paul Brinkman.

**Project administration:** Nele Albers, Willem-Paul Brinkman.

**Software:** Nele Albers.

**Supervision:** Mark A. Neerincx, Willem-Paul Brinkman.

**Validation:** Willem-Paul Brinkman.

**Visualization:** Nele Albers.

**Writing – original draft:** Nele Albers.

**Writing – review & editing:** Mark A. Neerincx, Willem-Paul Brinkman.

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
