## [Decision Letter · Decision Letter 0]

20 Jun 2022

PONE-D-22-03034Addressing people’s current and future states in a reinforcement learning algorithm for persuading to quit smoking and to be physically activePLOS ONE

Dear Dr. Albers,

Thank you for submitting your manuscript to PLOS ONE. After careful consideration, we feel that it has merit but does not fully meet PLOS ONE’s publication criteria as it currently stands. Therefore, we invite you to submit a revised version of the manuscript that addresses the points raised during the review process.

We look forward to receiving your revised manuscript.

Kind regards,

Fu Lee Wang

Academic Editor

PLOS ONE

Journal Requirements:

Reviewers' comments:

Reviewer's Responses to Questions

**Comments to the Author**

1. Is the manuscript technically sound, and do the data support the conclusions?

Reviewer #1: Yes

Reviewer #2: Partly

Reviewer #3: Yes

2. Has the statistical analysis been performed appropriately and rigorously? 

Reviewer #1: Yes

Reviewer #2: Yes

Reviewer #3: Yes

3. Have the authors made all data underlying the findings in their manuscript fully available?

Reviewer #1: Yes

Reviewer #2: Yes

Reviewer #3: Yes

4. Is the manuscript presented in an intelligible fashion and written in standard English?

Reviewer #1: Yes

Reviewer #2: Yes

Reviewer #3: Yes

5. Review Comments to the Author

Reviewer #1: The paper describes research that aims to investigate whether a reinforcement learning approach (taking into account current and future states) can be used to determine suitable persuasion types in order to persuade people to do preparatory activities for smoking cessation. The conducted study is a longitudinal study through which they compared 4 increasingly more complex versions of their algorithm. The reasoning behind algorithm components incorporates elements of the behavioural change field (such as Com-B) with persuasive technology approaches (such as the PSD model), which places the study at an intersection of two highly relevant fields.

The paper is clearly written and motivates well why certain factors are included and others are not. The steps taken are also relevant and well described. Furthermore, the data for this study is available at the indicated location and is neatly structured. The repository includes a README describing the contents, which also details how to perform the steps required to reproduce the data pre-processing and analysis. These steps are then further elaborated in similar README and manual files within the relevant subfolders. In addition, the authors also provide the code for the chatbot that was used in conducting the study.

All in all, the work appears to adhere to the 7 PLOS One criteria and investigates an important area of research since behaviour change applications still face major challenges with providing the right content at the right moment - e.g., leading to high dropout rates. Although the study's statistical analyses seem solid and clearly detailed, I must state that I'm not familiar enough with the specific Bayesian approach that was used to completely judge those steps thoroughly.

I include some more granular comments below.

Figures:

The text and colored lines in almost all Figures 2-4 are quite blurry, the authors might want to increase the resolution for these. A minor note is also that the color coding for numbers in Figures 2 and 4 might not be suitable for people who suffer from colorblindness. However, in practice this is mainly a concern for the indication of sessions in Figure 4 since for Figure 2 the discussion in the main text provides sufficient context to not need the colors.

Textual comments:

Line 12-13: "So Janine should also consider what type of person Janine is." -> 2nd Janine = Martha?

Line 69-79: "[..] how thoroughly people [?] to their activities." -> missing verb?

Line 79: The 'cold start problem' comes to mind, you touch upon this in your discussion, but might be worth a short elaboration.

Line 614: Perhaps add 'in the data repository'?

Finally, one minor comment regarding contents of the data repository:

The text in the codeblocks of a number of pdf files detailing how to perform the analyses runs outside of the codeblocks and often also the pages. E.g., in the Bayesian Comparison pdf file and the pdf files for the 1st Hypotheses.

Reviewer #2: While, in the submitted version of the article, the authors show substantial and impressive knowledge about the foundations and methods of social psychology (far beyond the assessment capabilities of this reviewer, to be honest) the account given of what they call ‘algorithms’ is definitely sketchy, to say the least, and insufficient to evaluate how the experiments were performed. More precisely, to this reviewer, the section ‘Methods’, as it stands, does not give an adequate description of the algorithmic technique(s) that the authors adopted in the study.

Below, I try to convey some more specific comments and suggestions for improvement that, to this reviewer, should be introduced in the due course of a major revision.

** Complexity

In theoretical computer science, the term ‘algorithm complexity’ designates very precise measurements: e.g, time complexity, memory complexity. In contrast, the authors use this term in a rather qualitative way, which I think should be better explained. The description of the so-called four levels of complexity of the algorithms are only accessible to the reader in Figure 2 and never explained in a satisfactory way. For clarity, this reviewer thinks that they should be summarized in a table, possibly within the main text and not in the appendix, and a precise methodological description should be given for each level.

** Algorithm

In the section ‘Methods’ and in the sub-section ‘Algorithm’ the authors characterize the overall RL framework in terms of a Markov Decision Process. Along this line, they describe the Action Space, State Space and Reward in a certain length (no objections here). What looks to be completely missing from that point on, however, is a description of the actual RL approach adopted. Below a list of a few remarkable doubts that remain unsolved:

1) The all-important MDP Transition Function is just mentioned but not described. The reader may conjecture that the authors implied that they could not know the MDP Transition Function beforehand, which would be reasonable, given the novelty of the approach. Is this the case?

2) In ‘complexity levels’ 3 and 4 (see Figure 2) the authors hint at the Q-value of each action, given a specific state. According to basic RL textbooks [1], the Q-value can be either computed beforehand, if the transition function is known in full, or learned from experience, like for instance with Q-learning, otherwise. Which approach is the one taken by the authors is unclear.

3) A text passage in page 8/26 reads as follows:

‘the virtual coach maintained a separate policy \\pi_i for each user i. When updating \\pi_i, an observed sample from user j was weighted based on how similar I and j were.’

which raises (at least) four sub-questions:

3.1) In RL the policy is the objective of learning. ‘One policy, multiple subjects’ could be a fitting description, by default. Now if the policy is per-subject, as the above text seems to imply, how is such policy learned? Typical RL optimization processes require a large number of episodes, not just one interaction, however protracted over time.

3.2) What is precisely a ‘sample’, to the authors? Is that the typical tuple <s, a=""> used in RL? And if so, with what method is policy \\pi_i updated?

3.3) If, in ‘complexity level’ 4, samples are collected not only for individual i but for a cohort of other individuals j, apart from the similarity measure (whose definition is given in the article), how is such evidence combined to have a policy in the sense of RL?

3.4) If samples from other individuals are used to update individual’s policy, how is time synchronization handled? In other terms, give that RL is incremental, subjects participating the experiments at later times may benefit from a larger quantity of samples collected from other subjects.

** Statistics

Once again the focus is on the algorithmic part. In fact, ‘complexity level’ 1 and 2 imply knowing the ‘best average reward’ and ‘best average reward in state’, respectively. How are these values computed? Are they determined during the random persuasion stages?

Other, minor suggested corrections and improvements:

** Abstract

- ‘To test the added value of subsequently incorporating these elements’

In the previous sentence we read "... variables such as ..." followed by some examples, but there is no specific term stating explicitly which elements the authors are talking about (Psychological dimensions? Past experiences?)

- ‘training the algorithm separately for the two types of activities may be more effective than training one single algorithm on the combined data set.’

Is it or is it not more effective? This is crucial to establish the contribution of the article.

** Text

Line 13: Possible typo ‘ Martha’

Line 73+: Such statement would deserve some reference in literature

Line 202+: The level of precision in defining the discount factor is somewhat surprising, given the overall account of the algorithm (see also above)

Line 264: Have the authors considered measures other that the Euclidean?

Line 435: How have entry errors been corrected?

Table 4: Being the mean perceived motivational impact computed on values which go from -10 to 10, the improvements obtained in this table are very low (The min and max values differ of 5.40 - 4.64 = 0.76, which represents a maximum improvement of 0.76/21 = 3.3%): is this correct? Such result should be further described (and better discussed).

Line 595: By looking at the measured moments in Tab.3, the statement doesn't seem evident. It could be useful to have a graph showing the evolution of the measures according to the moments.

[1] Richard S. Sutton and Andrew G. Barto, Reinforcement Learning: An Introduction, Second edition, in progress, https://web.stanford.edu/class/psych209/Readings/SuttonBartoIPRLBook2ndEd.pdf</s,>

Reviewer #3: This is a well-written paper with thorough description and evidence provided in attachments and a link where all the data and scripts are available. The authors apply Reinforcement Learning techniques to behavior change problems and there are still a lot of interesting challenges to tackle in this domain. The authors collected new data to examine whether different algorithm complexities would contribute to better performance of a reinforcement learning agent that was delivering persuasive messages to support people to change their physical activity or smoking behavior. The authors did a commendable job in providing clear stats of demographics within groups and to provide clear descriptions of their data and model in the readme files.

The research is novel and of interest, although the outcomes give us only limited insights.

I have some remarks about the contents of the paper, in search for more clarity and completeness of the work.

Major comments:

1. The research is based on two type of user behaviors: physical activity and smoking cessatation behavior. It was assumed that people could be persuaded in similar ways for those activities because 'they serve the same behavioral goal: quitting smoking.' This is a major assumption underlying the entire research and it seems flimsy and not sufficient. There are many behaviors that are very different but could contribute to the same goal. Moreover, smoking is addictive and as such I would expect at least that this would be taken into account for any persuasive strategy.

2. Algorithm description: the paper would benefit from a formal definition of all components of the algorithm, like action space and state space.

3. The persuasive messages seem not very personalized and quite general ('experts', 'people like you'). I would think it difficult to achieve true engagement from them. I do appreciate that the authors looked at other literautre to find templates that were validated, but would be interested in their opinion of how the content of their messages might have influenced the result.

4. The authors have only included self-report of motivation and effort for behavior as a measure of behavior change. This seems quite limited, in particular because the self-report was only for preperatory activities and not the actual behavior itself (smoking). Please include more information on what those type of activities are. It seems a limitation that the study does not evaluate whether the actual goal/target behavior was affected and the paper needs more argumentation on why this approach was chosen and even appropriate.

5. It seems the algorithm had very limited data to train on, given that it was on the data of just 2 sessions. So it has very limited information on which action would yield the best reward. Also, the study uses results of 521 participants, while 528 was required (and a 'conservative' number at that). In limitations is should be adressed how this could impact the conclusions and follow-up.

6. Please include results of how the agent was doing in terms of reward for the different groups and how this relates to the (involvement in) behavior that was done.

Minor comments:

1. The authors seem to generalize results to all behavior change applications from the start, where they assert that all applications assign their users activities. This seems a bold statement and does not link to any definition of behavior change or e-coaching solutions.

2. How do the interventions link to habit formation? Doing a certain prep behavior once or repeatedly should have different value.

3. It is unclear to what extent people were asked to do activities that they would normally not do, or that would be new to them. Also, did people that were included try to stop smoking before? This might impact their behevior.

4. Please include to what extent the message 'journey' indeed differed for participants in complexity 1-4.

5. Image 1 should be improved, it is unclear whether the actor is the human or the agent. Table 3 and 4 need more axes descriptions (not 1,2,3,4) and scale.

6. PLOS authors have the option to publish the peer review history of their article (what does this mean?). If published, this will include your full peer review and any attached files.

Reviewer #1: No

Reviewer #2: No

Reviewer #3: No

---

## [Author Response · Author response to Decision Letter 0]

12 Aug 2022

We would like to thank our academic editor and three anonymous reviewers for their detailed and helpful comments. We have addressed each reviewer comment to the best of our ability as described in the file "Response to Reviewers." We deliver the edited manuscript as well as a file that highlights the changes that we made to facilitate finding back the mentioned changes. We refer to line numbers in this file with tracked changes while describing how we addressed the reviewer comments.

---

## [Decision Letter · Decision Letter 1]

26 Sep 2022

PONE-D-22-03034R1Addressing people’s current and future states in a reinforcement learning algorithm for persuading to quit smoking and to be physically activePLOS ONE

Dear Dr. Albers,

Thank you for submitting your manuscript to PLOS ONE. After careful consideration, we feel that it has merit but does not fully meet PLOS ONE’s publication criteria as it currently stands. Therefore, we invite you to submit a revised version of the manuscript that addresses the points raised during the review process.

We look forward to receiving your revised manuscript.

Kind regards,

Fu Lee Wang

Academic Editor

PLOS ONE

Journal Requirements:

Reviewers' comments:

Reviewer's Responses to Questions

**Comments to the Author**

1. If the authors have adequately addressed your comments raised in a previous round of review and you feel that this manuscript is now acceptable for publication, you may indicate that here to bypass the “Comments to the Author” section, enter your conflict of interest statement in the “Confidential to Editor” section, and submit your "Accept" recommendation.

Reviewer #1: All comments have been addressed

Reviewer #2: All comments have been addressed

Reviewer #3: All comments have been addressed

2. Is the manuscript technically sound, and do the data support the conclusions?

Reviewer #1: Yes

Reviewer #2: Yes

Reviewer #3: Yes

3. Has the statistical analysis been performed appropriately and rigorously? 

Reviewer #1: Yes

Reviewer #2: Yes

Reviewer #3: Yes

4. Have the authors made all data underlying the findings in their manuscript fully available?

Reviewer #1: Yes

Reviewer #2: Yes

Reviewer #3: Yes

5. Is the manuscript presented in an intelligible fashion and written in standard English?

Reviewer #1: Yes

Reviewer #2: Yes

Reviewer #3: Yes

6. Review Comments to the Author

Reviewer #1: Thank you for the substantial revisions that have been made, not just regarding my comments and suggestions, but also regarding those of the other reviewers.

The revision has addressed al my aforementioned comments. I’ve also reviewed the authors’ responses to the other two reviewers and they seem to have made changes addressing their comments thoroughly as well, although I am not an expert on the specific type of algorithm employed by the authors. I therefore leave the final judgement on the changes regarding the other reviewers’ questions to those respective reviewers.

Reviewer #2: To this reviewer, revision R1 shows substantial progresses in the direction requested. In particular, the fundamental

algorithmic aspects of the approach proposed are described in a rather satisfactory way and the overall method can now be evaluated in full. As a word of caution, however, the authors should be aware that many of the algorithmic ideas that they adopt, Reinforcement Learning in the first place, are elaborated in a very specific way, to adapt these to the authors’ objectives. This is not a limitation ‘per se’ but calls for extra care in the description of the method proposed, to avoid generating confusion with readers being acquainted with more (allow me) ‘canonical’ versions of the techniques in point.

A few changes would still improve the readability of the article. Below I try to describe which aspects should be addressed as minor revisions.

** Algorithm description

Table 1 is a very welcome enhancement to the article, however, is literally plenty of aspects that cannot be understood in a first reading, since they require integrating other aspects that are clarified much further on. Also Table 2 is very useful, although it could be improved further.

As a matter of fact, I recommend to:

- put Table 2 ahead of Table 1 and modify the description of the algorithm in an incremental fashion;

- in Table 2 (will be Table 1) clarify how state space is defined, redirecting to text for better clarification

- in Table 2 add the abbreviations for the Action Space, which will be used in Table 1 (will be Table 2)

- in Table 2, both reward function and transition function are just defined as function types, which is not very informative: at least a minimal description should be given;

- Table 1 contains a lengthy and somehow confusing repeated definition of how states are defined: it would be advisable to extract the definition of states, possibly by making it a separate and smaller table, and then quoting just states as binary combinations in Table 1;

- in fact, the fundamental information content of Table 1 is the relation between states (whenever this applies) and the most effective actions, according to the metrics of choice.

** Overall Structure of Text

In the light of the comments above, the authors should consider that the contents of Table 1 seem to be pertaining more to a ‘Results’ section rather to a ‘Method’ one. Therefore, some re-arrangement of the overall description should be performed, to reflect this.

** One Final remark

The cited paper

Di Massimo F, Carfora V, Catellani P, Piastra M. Applying Psychology of Persuasion to Conversational Agents through Reinforcement Learning: an Exploratory Study. In: CEUR – Workshop Proceedings. vol. 2481; 2019.

has a journal version which I think should be cited in its place:

Carfora, V., Di Massimo, F., Rastelli, R., Catellani, P., & Piastra, M.. Dialogue management in conversational agents through psychology of persuasion and machine learning. Multimedia Tools and Applications, 39, 35949-35971; 2020.

Reviewer #3: The authors adressed the majority of my comments well. I would like to make to final remark for the authors to consider. It relates to my earlier comment (3.5), which is in essence about what constitutes success for this intervention. If you do an intervention which targets preperation for behavior, that preparation also contains behaviors that could be monitored (in this case for example watching a video and writing things down). To the reader it is not yet clear why self-report of effort is reliable as a (only) measure in this context (for instance, succesfull preparation could (and should) also translate to increased awareness or motivation, which was not assessed). Also, it would be nice to know how the authors aimed to make the self-report a more robust measure by mitigating reporting biases.

7. PLOS authors have the option to publish the peer review history of their article (what does this mean?). If published, this will include your full peer review and any attached files.

Reviewer #1: No

Reviewer #2: **Yes: **Marco Piastra

Reviewer #3: No

---

## [Author Response · Author response to Decision Letter 1]

3 Oct 2022

We very much thank our three reviewers for providing valuable feedback on our revised manuscript version. We describe how we have addressed the comments from the reviewers in the attached file called "Response to Reviewers."

---

## [Decision Letter · Decision Letter 2]

25 Oct 2022

Addressing people’s current and future states in a reinforcement learning algorithm for persuading to quit smoking and to be physically active

PONE-D-22-03034R2

Dear Dr. Albers,

We’re pleased to inform you that your manuscript has been judged scientifically suitable for publication and will be formally accepted for publication once it meets all outstanding technical requirements.

Kind regards,

Fu Lee Wang

Academic Editor

PLOS ONE

Additional Editor Comments (optional):

Reviewers' comments:

Reviewer's Responses to Questions

**Comments to the Author**

1. If the authors have adequately addressed your comments raised in a previous round of review and you feel that this manuscript is now acceptable for publication, you may indicate that here to bypass the “Comments to the Author” section, enter your conflict of interest statement in the “Confidential to Editor” section, and submit your "Accept" recommendation.

Reviewer #2: (No Response)

Reviewer #3: All comments have been addressed

2. Is the manuscript technically sound, and do the data support the conclusions?

Reviewer #2: (No Response)

Reviewer #3: Yes

3. Has the statistical analysis been performed appropriately and rigorously? 

Reviewer #2: (No Response)

Reviewer #3: Yes

4. Have the authors made all data underlying the findings in their manuscript fully available?

Reviewer #2: (No Response)

Reviewer #3: Yes

5. Is the manuscript presented in an intelligible fashion and written in standard English?

Reviewer #2: (No Response)

Reviewer #3: Yes

6. Review Comments to the Author

Reviewer #2: (No Response)

Reviewer #3: Thank you for adressing the concerns with vigour. I also appreciate the improved figures and descriptions created to adress the other reviewer's comments. In its current form I feel it is a nice addition to the literature for using RL in health behavior change.

7. PLOS authors have the option to publish the peer review history of their article (what does this mean?). If published, this will include your full peer review and any attached files.

Reviewer #2: **Yes: **Marco Piastra, PhD

Reviewer #3: No

---

## [Editor Report · Acceptance letter]

21 Nov 2022

PONE-D-22-03034R2 

Addressing people’s current and future states in a reinforcement learning algorithm for persuading to quit smoking and to be physically active 

Dear Dr. Albers:

I'm pleased to inform you that your manuscript has been deemed suitable for publication in PLOS ONE. Congratulations! Your manuscript is now with our production department. 

Kind regards, 

on behalf of

Professor Fu Lee Wang 

Academic Editor

PLOS ONE